# Closing the Gap in Proteomic Identification of *Histoplasma capsulatum*: A Case Report and Review of Literature

**DOI:** 10.3390/jof9101019

**Published:** 2023-10-15

**Authors:** Terenzio Cosio, Roberta Gaziano, Carla Fontana, Enrico Salvatore Pistoia, Rosalba Petruccelli, Marco Favaro, Francesca Pica, Silvia Minelli, Maria Cristina Bossa, Anna Altieri, Domenico Ombres, Nikkia Zarabian, Cartesio D’Agostini

**Affiliations:** 1Department of Experimental Medicine, University of Rome “Tor Vergata”, 00133 Rome, Italy; terenziocosio@gmail.com (T.C.); pistoiae@uniroma2.it (E.S.P.); favaro@uniroma2.it (M.F.); pica@uniroma2.it (F.P.); cartesio.dagostini@ptvonline.it (C.D.); 2Dermatologic Unit, Department of Systems Medicine, University of Rome Tor Vergata, Viale Oxford 81, 00133 Rome, Italy; 3Laboratory of Microbiology and Biological Bank, National Institute for Infectious Diseases Lazzaro Spallanzani IRCCS, 00149 Rome, Italy; carla.fontana@inmi.it; 4Laboratory of Clinical Microbiology, Policlinico Tor Vergata, 00133 Rome, Italy; rosalbapet81@gmail.com (R.P.); silvia.minelli@ptvonline.it (S.M.); mariacristina.bossa@ptvonline.it (M.C.B.); anna.altieri@ptvonline.it (A.A.); domenico.ombres@ptvonline.it (D.O.); 5School of Medicine and Health Sciences, George Washington University, 2300 I St NW, Washington, DC 20052, USA; nzarabian@gwmail.gwu.edu

**Keywords:** *Histoplasma capsulatum*, MALDI-TOF, reference database, literature review, proteomic, HIV, AIDS, fungi

## Abstract

Histoplasmosis is a globally distributed systemic infection caused by the dimorphic fungus *Histoplasma capsulatum* (*H. capsulatum*). This fungus can cause a wide spectrum of clinical manifestations, and the diagnosis of progressive disseminated histoplasmosis is often a challenge for clinicians. Although microscopy and culture remain the gold standard diagnostic tests for *Histoplasma* identification, matrix-assisted laser desorption ionization time of flight mass spectrometry (MALDI-TOF MS) has emerged as a method of microbial identification suitable for the confirmation of dimorphic fungi. However, to our knowledge, there are no entries for *H. capsulatum* spectra in most commercial databases. In this review, we describe the case of disseminated histoplasmosis in a patient living with HIV admitted to our university hospital that we failed to identify by the MALDI-TOF method due to the limited reference spectrum of the instrument database. Furthermore, we highlight the utility of molecular approaches, such as conventional polymerase chain reaction (PCR) and DNA sequencing, as alternative confirmatory tests to MALDI-TOF technology for identifying *H. capsulatum* from positive cultures. An overview of current evidence and limitations of MALDI-TOF-based characterization of *H. capsulatum* is also presented.

## 1. Introduction

### 1.1. Epidemiology

Histoplasmosis is a mycotic infection caused by *Histoplasma capsulatum*, a dimorphic fungus found globally but most prevalent in North and Central America [1]. It is common in the central and eastern regions of the United States, specifically in the Ohio and Mississippi River Valleys, but can also be found in parts of Central and South America, Africa, Asia, and Australia [2]. Two human variants of *H. capsulatum*, specifically *H. capsulatum* var. *capsulatum* (a New World human pathogen), which is the most common form, and *H. capsulatum* var. *duboisii* (an Old World human pathogen), which is more commonly found in Africa, have been described in humans [3,4]. A third variety was described in Africa, *Histoplasma capsulatum* var. *farciminosum* (an Old World animal pathogen), associated with epizootic lymphangitis in horses [3,4].

However, recent analyses of multiple isolates by whole genome sequencing recognized the existence of four phylogenetic species named *H. capsulatum* sensu stricto (s. str.) when referring to the Panamanian lineage (H81 lineage), *Histoplasma mississippiense* sp. nov. (formerly known as NAm 1), *Histoplasma ohiense* sp. nov. (formerly known as NAm 2), and *Histoplasma suramericanum* sp. nov. (formerly known as LAm A) [5].

### 1.2. Pathogenesis

This fungus is present in the environment, particularly in soil containing bird and bat guano [1], and grows in the environment in a hyphal form. When conidia or fragments of hyphae are inhaled by the host, they switch into a yeast form [6]. This morphologic transition is crucial for the ability of the fungus to cause disease in the host. The yeast forms are specialized for replication within host macrophages and travel from the lungs to the rest of the body. In the United States, *H. capsulatum* is the leading cause of endemic fungal respiratory infections [6]. This fungus can cause a wide spectrum of clinical manifestations, ranging from acute pulmonary infections to progressive disseminated forms involving one or multiple organs, and is often life-threatening [7]. The extent of disease is influenced by the infectious dose of inhaled conidia as well as by the host immune response and the integrity of the respiratory tract [8]. Individuals receiving immunosuppressive therapies or with untreated human immunodeficiency virus (HIV) require a small number of conidia to acquire a severe infection with widespread dissemination [9]. Since the yeast form of *H. capsulatum* replicates in macrophages, efficient host adaptive immunity and, specifically, the Th1 immune response are required to activate phagocytic cells and clear the infection [6]. In this regard, treatments with TNF-alpha inhibitors, which are commonly used among patients with inflammatory-autoimmune diseases, can impair the Th1 immune response, increasing the risk of disseminated histoplasmosis [10]. Additionally, biologics, i.e., IL-17 inhibitors and IL-12/23 inhibitors, which are commonly used among patients with inflammatory-autoimmune diseases such as psoriasis and inflammatory bowel diseases (IBD), may predispose to mycotic infections [11,12]. In fact, in addition to Th1, Th17 cells, through the production of IL-17, are crucial for protection against fungi by promoting the recruitment of neutrophils and macrophages at the site of infection [12]. In addition, patients living with HIV, especially those with CD4+ counts < 200 cells/μL [8], have a high risk of disseminated histoplasmosis. In these subjects, the shift of the Th1 to Th2 immune response to viral infection [13] and the impaired Th1 cell activity prevent the effective clearance of *H. capsulatum* infection. 

### 1.3. Diagnostic Tools

Histoplasmosis in immunocompromised individuals can occur with nonspecific symptoms, and the low index of suspicion can lead to diagnostic delay and increased mortality rates [14]. Thus, improvements in diagnostic methods are of great importance in the early and accurate detection of histoplasmosis, especially in immunocompromised patients, for prompt treatment. In the past few decades, various approaches have been developed for laboratory diagnostic accuracy of *Histoplasma* infection. Among these, *Histoplasma* antigen testing has greatly improved the laboratory diagnosis of histoplasmosis. *H. capsulatum* antigen testing detects polysaccharide antigens of the fungus and is usually performed on body fluids such as urine or serum samples [15]. Conflicting data are reported in the literature regarding the superiority of one of these assays over another, despite a few authors slightly favoring a urine sample over a serum sample due to high levels of *Histoplasma* antigenuria [16]. This diagnostic method allows for rapid diagnosis of probable histoplasmosis; however, the cross-reactivity with other dimorphic fungi antigens, such as *Paracoccidioides brasiliensis, Blastomyces dermatitidis, Coccidioides immitis,* and *Penicillium marneffei,* represents a critical limitation of this test [17]. Furthermore, *Histoplasma* antigen detection testing has proven to be most useful for the diagnosis of disseminated histoplasmosis, moderately useful for localized pulmonary histoplasmosis, and not useful for remote diseases [18]. *H. capsulatum* antibody testing is another diagnostic method that has become increasingly popular among many laboratories due to the convenience, availability, and accuracy of these assays. The three most common serologic assays for histoplasmosis include the immunodiffusion (ID) test, complement fixation (CF) test, and enzyme immunoassay (EIA). However, these tests are mostly utilized in chronic histoplasmosis infections since they show low sensitivity, principally in acute forms of the disease. Moreover, they should not be used in patients with the disseminated form of histoplasmosis because of an increase in false-negative results [19]. The low sensitivity of the antibody detection tests and the fact that the antibodies are detectable around 4–8 weeks after initial infection with acute pulmonary histoplasmosis represent the major limitations, especially among immunocompromised individuals unable to generate an effective immune response to *H. capsulatum* antigens [19]. Molecular “in-house” methods of *H. capsulatum* identification from clinical specimens or cultures are available at some research institutions [20,21]. Despite the lack of an accessible commercial database for the standardized molecular identification of fungi, molecular approaches represent a valuable tool for the laboratory diagnosis of *H. capsulatum*. Dantas et al. [22], by comparing mycological, serological, and molecular methods for the diagnosis of histoplasmosis in patients with or without HIV, found that molecular techniques may provide accurate identification of *Histoplasma* even if serological and mycological tests are negative. However, according to Stempak et al. [23], these methods must not be considered a substitute for culture but rather an adjunct that can be useful when fungi are detected in an anatomic pathology specimen but fail to grow in culture or a corresponding specimen is not submitted for cultural tests.

Culture and histopathological analyses, showing the characteristic intracellular forms of the fungus in infected tissues, are considered, to date, the gold standard for the diagnosis of histoplasmosis [10]. Unfortunately, *H. capsulatum* may be misidentified in histologic sections as *Candida glabrata* and *Sporothrix schenckii* due to the variety of morphologically similar small yeast forms. Oftentimes, cultures are limited by the slow growth of the fungus and require additional tests to confirm the diagnosis [19]. In recent years, MALDI-TOF MS has been established as a valuable microbial confirmation tool [24]. This method uses proteomics, including the structure, function, and interactions of proteins [25], to confirm the diagnosis of microbial isolates. However, MALDI-TOF efficiency depends mainly on the depth of reference databases, and, to our knowledge, there are no entries for *H. capsulatum* spectra in the commercial Bruker Daltonics databases until today. Although the MALDI-TOF MS method in the clinical microbiology laboratory allows for a revolutionizing infectious disease diagnosis, herein we present the case of HIV-associated histoplasmosis that we were not able to diagnose microbiologically by the MALDI-TOF method due to the absence of the reference spectrum in the instrument database. Furthermore, herein we provide a review of the literature focusing on the application of the MALDI-TOF method for the identification of *H. capsulatum*, highlighting the strengths and limitations of this proteomic-based technology.

## 2. Case Report

A 46-year-old HIV-positive Italian man, living between Italy and Argentina, was admitted to our university hospital with suspected pneumonia, suffering from the following symptoms: cough, fever, myalgia, and asthenia for the last seven days. Laboratory tests showed leukopenia (leukocyte count of 2990/mm^3^), with a low number of LTCD4+ (16 cells/mm^3^) and a CD4/CD8 ratio of 0.4, anemia (red blood cells count of 3 × 10^6^/μL, hemoglobin 8.7 g/dL, and hematocrit 25.7%), and an increase in inflammatory markers (procalcitonin 0.71 ng/mL and C-reactive protein 211.11 mg/L). Based on a clinical suspicion of a lower respiratory tract infection, bronchoalveolar lavage fluid (BAL) was collected to perform conventional microbiological tests, including bacterial and fungal culture tests. PCR assays for *Mycobacterium tuberculosis*, *Mycoplasma pneumoniae*, *Haemophilus influenzae*, *Streptococcus pneumoniae,* and intracellular bacteria (*Legionella pneumophila, Chlamydophila pneumoniae*) detection as well as the FILMARRAY^TM^ Respiratory Panel for viral and bacterial identification were also performed.

## 3. Materials and Methods

### 3.1. Microbiological Culture and Microscopy

The BAL sample was inoculated onto two sets of Sabouraud Dextrose agar (SDA) (Difco Laboratories, Detroit, MI, USA), with and without cycloheximide, and incubated at 25 °C and 37 °C for fungal identification and onto Chocolate Blood agar (CBA), MacConkey agar, and Columbia CNA agar for bacterial identification. The colony morphology of positive cultures was analyzed by a light microscope (Olympus, Carl Zeiss, UK) with 20× and 100× magnification.

### 3.2. MALDI-TOF Method

The acquisition and analysis of mass spectra were performed by a MALDI Biotyper^®^ (MBT) (Bruker Daltonics, Bremen, Germany) using the MALDI Biotyper software package (version 3.4) with the Filamentous Fungi Library 4.0 (Bruker Daltonics, Bremen, Germany) and default parameter settings as published previously [26]. The Bruker bacterial test standard (Bruker Daltonics, Bremen, Germany) was used for calibration according to the instructions of the manufacturer. Briefly, a part of the colony of the suspected fungus, in the yeast form, was gently scraped with sterile plastic pliers, transferred into an Eppendorf tube containing 1 mL of HLPC-grade water, vigorously mixed for one minute, and then centrifuged for 2 min at 13,000 rpm. The sample was first suspended in 75% ethanol HPLC (Sigma-Aldrich, Milan, Italy) and vigorously mixed. Then, the hydro-alcoholic solution was removed by centrifugation at 13,000 g for 2 min. Next, the pellet was suspended in 20 μL of 70% formic acid (Sigma-Aldrich, Milan, Italy) by vigorously pipetting the sample up and down. After a 5-min incubation, 20 μL of acetonitrile HPLC (Sigma-Aldrich, Milan, Italy) were added. The suspension was vigorously mixed and centrifuged for 2 min at 13,000 rpm. One microliter of the supernatant was deposited in 4 replicates on a polished steel target (MTP384, Bruker Daltonics GmbH, Bremen, Germany) and air-dried. To avoid cross-contamination, a new pipette tip was used for each sample position. Each deposit was then covered with 1 μL of a freshly prepared solution of α-cyano-4- hydroxycinnamic acid (HCCA) (Bruker Daltonics GmbH, Bremen, Germany) in 50% acetonitrile HPLC (Sigma-Aldrich, Milan, Italy) and 2.5% trifluoroacetic acid HPLC (TFA) matrix. The Biotyper software compares each sample mass spectrum to the reference mass spectra in the database, calculates an arbitrary unit score value between 0 and 3, reflecting the similarity between the sample and reference spectrum, and displays the top 10 matching database records. Standard Bruker interpretative criteria were applied. Briefly, scores of ≥ 2.0 were accepted for species assignment, and scores of ≥ 1.7 but < 2.0 were accepted for identification at the genus level. Scores below 1.7 were considered unreliable. In addition, consistency categories A (species consistency), B (genus consistency), and C (no consistency), which are assigned to the identifications by the Biotyper software, were considered for identification. Variations in the cutoff score value were carried out by reducing the species cutoff value to 1.9, 1.8, and 1.7 and the genus cutoff value to 1.6 and 1.5, followed by reinterpreting the top 10 matching database records.

### 3.3. PCR Amplification and DNA Sequencing

DNA was extracted from positive cultures suspected to be *H. capsulatum* using the MOLgen Universal Extraction Kit (ME 188830, Adaltis, Italy) following the manufacturer’s instructions. Nucleic acid was amplified using primers P1 and P2, which amplify the Large Subunit (LSU) variable region on the 28S gene [27], according to the following protocol: taq DNApolymerase 2x (pcrBio) 10 µL, primer mix P1 and P2 [10 picomoles/µL], each 2 µL, extracted sample 1 µL, H_2_O 7 µL, and amplified with the 9700 PCR systems (Applied Biosystems, Foster City, CA, USA). After the amplification, the samples were purified by the QIAquick PCR purification kit Qiagen (Qiagen, Hilden, Germany) and sequenced using the Brilliant DYE Terminators cycle sequencing kit v 1.1 (BRD1-100)(NimaGen Lagelandseweg 56 6545 CG Nijmegen-The Netherlands). The sequences were analyzed by ABI Prism 310 (Applied Biosystems, Foster City, CA, USA). Primers P1 ATCAATAAGCGGAGGAAAAG and P2 CTCTGGCTTCACCCTAATC were used for both amplification and sequencing. The electropherogram was analyzed by “Sequence Analysis” Version 5, and the sequence was analyzed by the NCBI BLAST program.

### 3.4. Search Strategy

We performed a comprehensive search of the following databases from 2015 to June 2023: Cochrane Central Register of Controlled Trials; MEDLINE; Embase; US National Institutes of Health Ongoing Trials Register; NIHR Clinical Research Network Portfolio Database; and the World Health Organization International Clinical Trials Registry Platform. Reference lists and published systematic review articles were analyzed. We used the term “*Histoplasma*” with the following keywords, separately and in combination: “MALDI”, “MALDI TOF”, “MALDI TOF MS”, and “VITEK MS”. Only English-language articles were included in the search. Forward citation analyses of original studies and review articles were also conducted.

### 3.5. Inclusion Criteria

In all the studies in which, in addition to MALDI, other proteomic approaches were used for *H. capsulatum* identification, we considered only the results related to MALDI analysis. All human studies were included with no age, sex, ethnicity, or type of scientific study-related restriction. We considered only case reports and case series in which MALDI was used for *H. capsulatum* identification and all case reports that have not yet been included in reviews or clinical trials.

### 3.6. Exclusion Criteria

Exclusion criteria were articles discussing proteomic methods other than MALDI-TOF and non-English-language publications.

## 4. Results

### 4.1. Culture-Based and Microscopy Identification of H. capsulatum from a Bronchoalveolar Lavage Sample

All the laboratory tests performed for bacterial and viral pathogen identification were negative, except for the fungal cultures. In fact, these latter showed the presence of colonies that were slow-growing (15 days), white to buff-brown colored, suede-like to cottony with a pale brown reverse at 25 °C, or smooth, moist, and yeast-like at 37 °C. Microscopy analysis (Figure 1) of fungal cultures at 25 °C showed the presence of large, rounded, single-celled tuberculate macroconidia formed on short, hyaline, undifferentiated conidiophores that are a hallmark of *H. capsulatum.* Small, round to pyriform microconidia borne on short branches on the sides of the hyphae were also present. Based on the microscopic features, a differential diagnosis with various fungal molds, including *Sepedonium* and *Chrysosporium* spp., has been made by performing subcultures at 37 °C to demonstrate the conversion from the mold form to the yeast form that is necessary to positively identify dimorphic pathogens as *H. capsulatum*.

### 4.2. MALDI-TOF MS-Based Identification of H. capsulatum

The proteomic analysis of the fungus demonstrated that MALDI-TOF misidentified *H. capsulatum* as other fungal pathogens and, specifically, as *Aspergillus niger*, *Chrysosporium shanxiense, Microsporum audouinii,* and *Talaromyces rugulosus* (Appendix A) due to the lack of an *H. capsulatum* reference profile in the instrument database. In fact, the low score values, ranging from 1.15 to 1.05, clearly suggest the absence of reliable identification at the genus level. Thus, based on the study results, we can assume that Bruker MALDI-TOF technology is not a valuable tool for detecting *Histoplasma* in positive cultures. Thus, the laboratories can either implement Bruker databases with the updated filamentous fungal library once it is approved or construct and validate a *Histoplasma* library themselves, as Valero et al. [28] did for a correct identification of the fungus.

### 4.3. Identification of H. capsulatum by Sequencing PCR-Amplified DNA

To definitively confirm the laboratory diagnosis of histoplasmosis, the DNA from a fungal culture was extracted and characterized by sequencing the LSU regions of the 28S gene. The results from the sequencing analysis aligned with BLAST. NCBI (https://blast.ncbi.nlm.nih.gov/Blast.cgi (accessed on 2 October 2023)) (Appendix A) confirmed the suspected culture isolate as *H. capsulatum* var. *capsulatum*. The results of the sequencing have been deposited in the GeneBank with the following accession number: OR604568.

### 4.4. Literature Review of MALDI-TOF-Based Identification of H. capsulatum

Twelve articles reporting the application of the MALDI-TOF method for *H. capsulatum* identification were included in the analysis. Of these, four were excluded after the application of the exclusion criteria. Among the eight remaining publications eligible for analysis, two were excluded after reading the abstract or the full text. Only six articles were included in the study (Appendix A). The results of our research are summarized in Table 1.

## 5. Discussion

Proteomics is a challenging technique in the identification of microbial species, and in particular, MALDI-TOF MS is one of the most commonly used proteomic techniques to structurally identify proteins and peptides in clinical microbiology [34]. As with other automated microbial identification systems, MALDI-TOF MS also relies on a reference database for the identification of bacteria and fungi. Although the instrument manufacturers follow similar principles, the major differences depend on the procedure and algorithm used in creating their own reference databases. While in areas where histoplasmosis is considered an endemic infection, MALDI-TOF technology can be used to rapidly identify *H. capsulatum*, in non-endemic countries, MALDI-TOF-based identification of *Histoplasma* remains difficult due to the lack of entries for *H. capsulatum* spectra in most commercial databases. Here, we report the case of *H. capsulatum* infection in a patient living with HIV where Bruker MALDI-TOF technology (Bruker Daltonics) failed to identify the fungus, highlighting the fact that MALDI-TOF reliability depends mainly on the accuracy and breadth of commercially available databases [35]. Our literature search found six publications aimed at evaluating the accuracy of MALDI technology in *Histoplasma* identification, addressing the strengths and limitations of this laboratory diagnostic method. In detail, Panda et al. [29] carried out a study on 125 fungal isolates (88 yeasts and 37 molds) in order to identify the strains at the genus and species level using MALDI-TOF technology. As in our study, they used the Biotyper 3.1 software (Bruker Daltonics, Bremen, Germany)). The instrument, despite including more than 500 references for fungi, was not able to identify *H. capsulatum* due to the lack of the fungal reference spectrum in its database. The creation of an in-house database, supplementing the limited commercial databases, has been proven to be highly advantageous for the identification of *H. capsulatum.* In this regard, Valero et al. [28] created an “in-house” reference database to test 30 *H. capsulatum* isolates from the Collection of the Spanish National Centre for Microbiology. The manufacturer defines an MS score between 1.7 and 2.0 as reliable genus-only identification. In a blind assay, all 30 isolates studied were correctly identified with a score above 1.70 (86.6%). Furthermore, the authors [28] developed the first reference mass spectrometry database for the identification of *H. capsulatum* in both mycelium and yeast growth phases using MALDI-TOF technology, with the most reliable results for the mycelial phase. In contrast to Bruker Daltonics, the Vitek MS v3.0 database is a reference database that encompasses 1046 clinically significant organisms, including *Histoplasma capsulatum* [34]. Rychert et al. [30], using the Vitek MS v3.0 database, evaluated 141 isolates of dimorphic fungi, including *Blastomyces dermatitidis*, *Coccidioides immitis*/*posadasii, Histoplasma capsulatum*, and the *Sporothrix schenckii* complex, as part of FDA trial testing. Specifically, the study included 40 *Blastomyces dermatitidis* isolates, 38 *Coccidioides immitis/posadasii* isolates, 32 *Histoplasma capsulatum* isolates, and 31 *Sporothrix schenckii* complex isolates. The Vitek MS v3.0 was able to identify all 141 dimorphic fungi tested, including *H. capsulatum*. In other studies, MALDI-TOF MS technology has been found to be particularly valuable in the diagnosis of *H. capsulatum* in immunocompromised patients, for whom a rapid diagnosis is essential for prompt treatment. In particular, Le et al. [31] reported a case of *H. capsulatum* infection in an immunocompromised patient that they could diagnose, after weeks of failed diagnostic examinations, using MALDI-TOF MS technology. Furthermore, as reported by Velayuthan et al. [32], MALDI-TOF MS analysis allowed for a confirmed diagnosis of *Histoplasma* infection in an HIV-positive Indonesian patient suspected of being infected by the opportunistic fungus *Talaromyces marneffei,* a fungus that is endemic in Malaysia. In another work, Montagnac et al. [33] reported a case of histoplasmosis that was diagnosed by MALDI-TOF and PCR assay in a patient whose symptoms resembled a lung carcinoma. Overall, these studies suggest that MALDI-TOF represents a promising tool for the rapid and efficient diagnosis of fungal infections. However, to date, the literature data and our laboratory experience clearly show that MALDI-TOF efficiency depends mainly on the width of the reference database. Thus, only continuous update and improvement of the instrument databases, by expanding them with fungal strains derived from routine sample cultures, may improve the efficiency of MALDI-based identification of dimorphic fungi [36]. Our study also demonstrates that the PCR-based approach, in association with conventional diagnostic methods (i.e., culture and histopathology), can be considered an alternative and valuable tool to the MALDI method for confirmation of culture isolates suspected to be *H. capsulatum.* Several factors influence the reliability and reproducibility of molecular tests, including the type of samples tested. Fungal culture represents the best-performing sample, but this has little applicability in the routine diagnostic laboratory since the fungus can take a long time, 4 to 6 weeks, to grow before PCR application [37]. The whole blood and serum are also appropriate specimens for PCR testing, with very high sensitivity and specificity, whereas molecular diagnosis can be applied to lung biopsies when other classical diagnostic methods used for *Histoplasma* identification are usually not effective [20]. Although many of the PCR tests for *Histoplasma* identification were developed in-house and a meta-analysis study showed a lack of consensus regarding the methodologies, particularly PCR protocols and gene targets (100-KDa-like protein genes and amplification of ITS regions), molecular tests have been shown to have a good clinical performance in patients with histoplasmosis [20].

As reported by Martagon-Villamil et al. [38], the PCR assay has been shown to be 100% sensitive and 100% specific for the differentiation of *H. capsulatum* from other genetically related fungi, including *B. dermatitidis*, or with fungi that have forms that resemble those of *C. glabrata*, *S. schenckii,* or *T. marneffei.* Most studies reported in the literature have been based on nested PCR, and more recently, real-time quantitative PCR (qPCR) methods have been developed for human diagnosis to overcome the limits of the microbiological culture [20]. To improve the sensitivity of existing real-time quantitative PCR (qPCR) assays, Alanio et al. [4] further developed a new RT-qPCR assay that allows amplification of whole nucleic acids (WNA) instead of DNA and ribosomal small subunit RNA (*mtSSU)* instead of *ITS* as the target gene of *Histoplasma* spp. This test performed better in the detection of *H. capsulatum var. duboisii* than the other conventional diagnostic methods [4]. This is of great clinical importance considering that histoplasmosis due to *H. capsulatum var. duboisii* is frequently underdiagnosed in Africa. Furthermore, a highly sensitive assay on blood can help with the post-treatment follow-up of patients with histoplasmosis.

Thus, to date, molecular approaches may be considered a promising laboratory tool to identify *H. capsulatum* from clinical specimens or cultures in all countries where histoplasmosis infection is not endemic. 

Moreover, we need to consider the impact of climate change on the growing incidence of specific fungal infections. The burden posed by these infections might worsen due to the more frequent occurrence of extreme weather events. Specifically, the ongoing rise in temperatures, especially in higher latitudes, plays a role in the gradual expansion of the geographic ranges of dimorphic fungi such as *Coccidioides*, *Blastomyces*, *Histoplasma*, and *Sporothrix*. Hence, there is an urgent and unmet need to address the enhancement of the diagnostic pathway [39,40].

## 6. Conclusions and Perspectives

Although culture is the current gold standard for laboratory diagnosis of histoplasmosis, this method is a time-consuming process and can provide unreliable sensitivity and specificity. Rapid diagnosis and treatment of histoplasmosis is crucial to decreasing mortality, especially among immunocompromised patients such as those with untreated HIV and AIDS. By chance, the lack of identification of *H. capsulatum* by the MALDI method did not affect our patient’s outcome because the growth of the fungus in the filamentous form at 30 °C led clinicians to start an empirical therapy based on liposomal amphotericin B, which, as reported by the guidelines, is the first-line antifungal drug in patients with HIV [41]. Technological advancements have allowed for the emergence of rapid and accurate diagnostic methods for histoplasmosis using antibodies, antigens, and molecular tests. However, both antigen and antibody detection methods pose some limitations. Antigen testing often cross-reacts with other dimorphic fungal antigens, providing inaccurate results, while antibody testing has low sensitivity, especially in immunocompromised individuals, can cross-react, and it takes weeks for antibodies to arise, so it is not as helpful as a diagnostic. In recent years, the emergence of proteomics, specifically MALDI-TOF MS technology, has greatly improved the diagnostic accuracy of *H. capsulatum*. However, its accuracy mainly depends on the availability of the *H. capsulatum* microbial isolate in the proteome spectra database. Therefore, continuous expansion of the spectral database, alongside the development of a well-curated open-access mass spectra database, may improve the efficiency of the MALDI-TOF technique in identifying dimorphic fungi, including *H. capsulatum,* in those countries where these pathogens are not considered endemic. As an alternative to the MALDI-TOF method, the most common molecular tests, based on PCR amplification and sequencing, have been shown to possess the potential to facilitate the diagnostic routine of histoplasmosis and are arguably important, especially in those regions where the infection is not endemic, due to limited expertise in *H. capsulatum* identification. Furthermore, due to the growing burden of fungal infections and the recent rise of antifungal drug resistance in fungi, antifungal susceptibility testing (AFST) is of increasing importance. The common methods of AFST have turnaround times of 24 to 48 hours, and the available rapid methods are limited by applicability, cost efficiency, or accuracy. There are currently no standardized methods for susceptibility testing for the dimorphs, and there are no breakpoints.

In addition to allowing rapid and reliable identification of bacteria and fungi, several approaches have been proposed for MALDI-TOF MS application to AFST [42,43]. Knoll et al. [44] reported that MALDI-TOF MS could be a promising new method for the rapid detection of resistance in fungi, with a sensitivity of 91% and a specificity of 95% overall. However, none of the studies evaluated by Knoll et al. [44] included clinical isolates of *H. capsulatum*. In this context, dimorphic fungi, such as *H. capsulatum,* show increasing resistance to echinocandins and fluconazole, causing treatment failure in HIV patients [45]. Hence, further studies should be focused on MALDI-TOF MS-based AFST as a rapid and reliable tool for the detection of drug resistance in *H. capsulatum* clinical isolates in order to ensure prompt and appropriate treatment for HIV-positive patients with histoplasmosis.

## Figures and Tables

**Figure 1 jof-09-01019-f001:**
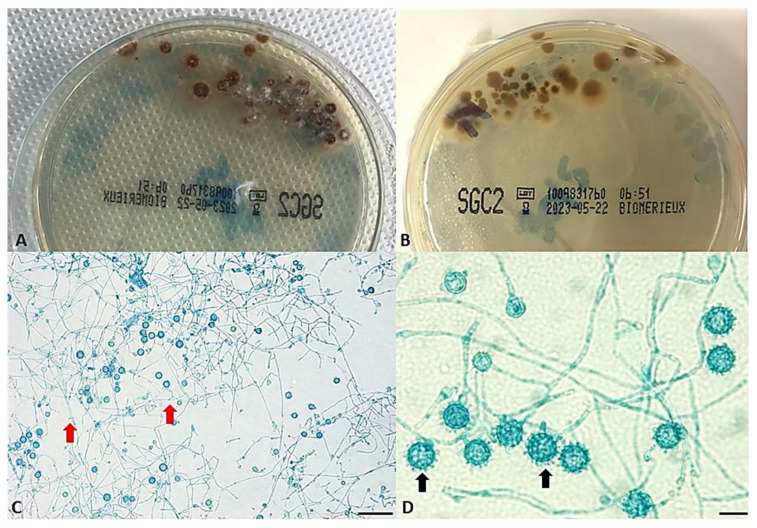
(**A**,**B**) The colonies on SDA at 25 °C showed white to buff-brown colored, suede-like to cottony mycelium on the recto and a pale brown reverse. (**C**) Microscopy analysis shows the presence of small, round to pyriform microconidia born on short branches on the sides of the hyphae at 20× (red arrows), scale bar 50 µm, and (**D**) characteristic large, rounded, single-celled, tuberculate macroconidia (black arrows) formed on short, hyaline, undifferentiated conidiophores at 100×, scale bar 10 µm.

**Table 1 jof-09-01019-t001:** Results of research in the literature.

	MALDI Database Version	Numbers of Strains	MALDI Score Identification	Phase ^b^
Panda et al. 2015 [29]	MALDI Biotyper 3.1 database, Bruker Daltonics	1	NA- *H. capsulatum* was not available in the database	NA
Valero et al. 2018 [28]	MALDI Biotyper Database v.4.0.0.1,	6	1.78 ± 0.15	3 M
	Bruker Daltonics			3 M/Y
Rychert et al. 2018 [30]	Vitek MS v3.0	32	Over 90% correct identification at the genus level	NA
Le et al. 2021 [31]	NA ^a^	1	NA	NA
Velayuthan et al. 2020 [32]	NA ^a^	1	NA	NA
Montagnac et al. 2019 [33]	NA^a^	1	NA	NA

^a^ *H. capsulatum* is included in one FDA-cleared database (Vitek MS) but was not specified in the manuscript. ^b^ Both morphological phases, M(mycelium) and Y(yeast).

## Data Availability

Data sharing is not applicable.

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
