# Peer review of "Closing the Gap in Proteomic Identification of Histoplasma capsulatum: A Case Report and Review of Literature"

_jof, 2023, doi:10.3390/jof9101019_

Round 1

Reviewer 1 Report

In the manuscript titled “Closing the gap in proteomic identification of dimorphic fungi: a case report of Histoplasma capsulatum infection and review of literature”, authors Cosio et al present a case of histoplasmosis in a HIV-positive Italian patient and analyze the current literature available for Histoplasma sp. identified by MALDI-ToF. Overall the authors recognize and emphasize the need for better diagnostics for histoplasmosis, especially in non-endemic regions.

I believe this manuscript is well suited for the Special Issue “Histoplasma and Histoplasmosis 2023” so long as the authors make a few edits to strengthen their review.

MALDI-ToF has become the workhorse of many clinical labs across the globe as it’s ease, time-to-results, and non-targeted approach have significantly improved microbiology diagnostics. As the authors point out, this is a common way diagnostic labs will identify pathogens, however, there still are some caveats to its use and it is not the only diagnostic tool we can use for a diagnosis.

In order to strengthen this review, I suggest the following:

-The authors claim there are no entries of Histoplasma in most databases but this is not true and the authors contradict this statement later in the manuscript as well (line 278). Though Histoplasma is not currently a part of the Bruker library, it has been a part of the Vitek database for years and Bruker is going to be adding it to their new updated filamentous fungi library soon. This should be clarified.

- The introduction should be broken up into paragraphs to follow better. Authors should also expand this section by talking more in depth about other diagnostic tests available as well as their sensitivities, specificities, caveats, and benefits (i.e. time to ID, whether it’s a targeted approach or not, cost, etc). I direct the authors to the review by Zhang et al. (J Clin Microbiol. 2021 Jul; 59(7): e01784-20.) which summarizes current diagnostics well.

-Clarify the dates of the literature search. Why was 2018 used as a start date but a paper from 2015 was included?

-For the case report, authors can expand on how exactly the lack of MALDI ID affected the patient. How long was the ID delayed? Was their length of stay affected? Was treatment affected? Be specific. 

-Reword to clarify when pictures of culture growth and microscopic pictures were taken. It may have taken 15 days for colonies to appear but were these pictures taken on day 15 or later? Could you also include a picture of the converted yeast form?

- The antifungal testing methods mentioned (lines 340 – 341, CLSI M27) are actually for Candida testing, not filamentous or dimorphic fungi. There are currently no standardized methods for susceptibility testing for the dimorphs and there are no breakpoints (see M38).

Minor comments:

-Line 49: Clarify that histoplasmosis is the leading cause of endemic fungal respiratory infections.

-Typically MALDI libraries as described by the “depth” of their library rather than their “width”

-Line 123: Filmarray respiratory panel also includes bacterial targets

-Lines 166-171: Add the gene targeted by the P1 and P2 primers

-Add a scale bar to microscopic pictures in Figure 1 and change “yellow arrows” in description to “red”

-Line 228: reword as labs can either update their Bruker databases with the updated filamentous fungal library once it is approved, or they can construct and validate a Histoplasma library themselves as Valero et. al did.

-Lines 238-240: Be specific why 6 out of the 12 articles you found were excluded.

-Line 326: Expand. Antibody testing can also cross react and it takes weeks for antibodies to arise so it’s not as helpful as a diagnostic.

-Figure S1: Label each spectra and add the Histoplasma spectra as reference.

-Figure S2: show the subject ID. Can include a screenshot of the blast results with the percent identification

-Figure S3: Don’t use. This is for Candida, not Histoplasma and cannot be adapted

Minor grammatical and spelling errors must be addressed. Overall English is alright. 

Author Response

Dear Editor, 

      Thank you very much for giving us the opportunity to revise our manuscript (Submission ID: 2595986). We also would like to thank the reviewers for their valuable comments that undoubtedly will improve the quality of our paper.

Here, we wrote a point-to-point letter in order to respond to the reviewer’s comments. All the changes made are in red font (mark copy).

In the manuscript titled “Closing the gap in proteomic identification of dimorphic fungi: a case report of Histoplasma capsulatum infection and review of literature”, authors Cosio et al present a case of histoplasmosis in a HIV-positive Italian patient and analyze the current literature available for Histoplasma sp. identified by MALDI-ToF. Overall the authors recognize and emphasize the need for better diagnostics for histoplasmosis, especially in non-endemic regions.

I believe this manuscript is well suited for the Special Issue “Histoplasma and Histoplasmosis 2023” so long as the authors make a few edits to strengthen their review.

MALDI-ToF has become the workhorse of many clinical labs across the globe as it’s ease, time-to-results, and non-targeted approach have significantly improved microbiology diagnostics. As the authors point out, this is a common way diagnostic labs will identify pathogens, however, there still are some caveats to its use and it is not the only diagnostic tool we can use for a diagnosis.

  1. R. Dear Reviewer, thank you for your interest in our manuscript and useful comments.

In order to strengthen this review, I suggest the following:

  • The authors claim there are no entries of Histoplasmain most databases but this is not true and the authors contradict this statement later in the manuscript as well (line 278). Though Histoplasma is not currently a part of the Bruker library, it has been a part of the Vitek database for years and Bruker is going to be adding it to their new updated filamentous fungi library soon. This should be clarified.

A.R. Dear reviewer thank you for your suggestion. We have clarified and corrected the manuscript according to your comments.

However, MALDI-TOF efficiency depends mainly on depth of reference databases, and, to our knowledge, there are no entries of H. capsulatum spectra in the commercial Bruker Daltonics databases until today”.

  • The introduction should be broken up into paragraphs to follow better. Authors should also expand this section by talking more in depth about other diagnostic tests available as well as their sensitivities, specificities, caveats, and benefits (i.e. time to ID, whether it’s a targeted approach or not, cost, etc). I direct the authors to the review by Zhang et al. (J Clin Microbiol. 2021 Jul; 59(7): e01784-20.) which summarizes current diagnostics well.

A.R. Dear reviewer, thank you for your suggestion. We have modified the main text according to your and Reviewer 2 suggestions. Moreover, we have read and cited the review by Zhang et al. (J Clin Microbiol. 2021 Jul; 59(7): e01784-20.).

  • Clarify the dates of the literature search. Why was 2018 used as a start date but a paper from 2015 was included?

A.R. We have corrected the dates of the literature search as follows:

We performed a comprehensive search of the following databases from 2015 to June 2023: Cochrane Central Register of Controlled Trials; MEDLINE; Embase; US National Institutes of Health Ongoing Trials Register; NIHR Clinical Research Network Portfolio Database; and the World Health Organization International Clinical Trials Registry Platform. Reference lists and published systematic review articles were analyzed. We used the term “Histoplasma” with the following keywords, separately and in combination: “MALDI”, “MALDI TOF”, “MALDI TOF MS”, “VITEK MS”. Only English language articles were included in the search. Forward citation analysis of original studies and review articles were also conducted.”

  • For the case report, authors can expand on how exactly the lack of MALDI ID affected the patient. How long was the ID delayed? Was their length of stay affected? Was treatment affected? Be specific. 

A.R. Dear reviewer thank you for your suggestion. The failure identification of Histoplasma capsulatum by MALDI did not allow us to provide a definite diagnosis of histoplasmosis. The fungal structures visualized microscopically resembled those of H. capsulatum but they can be confused with structures from other fungal pathogens such as Sepedonium spp. or Chrysosporium spp. .  However, the lack of identification of H. capsulatum by MALDI method did not affect the patient’s treatment because the growth of the fungus in the filamentous form at 30°C led clinicians to start an empirical therapy based on liposomal amphotericin B that, as reported by the guidelines, is the first line antifungal drug in patients with HIV.  The therapy with amphotericin B was maintained even after the correct identification by PCR. Regarding the clinical case report, we want to underline that this article has focused on the laboratory diagnosis of histoplasmosis rather than on the clinical and therapeutic management of the patient.

  • Reword to clarify when pictures of culture growth and microscopic pictures were taken. It may have taken 15 days for colonies to appear but were these pictures taken on day 15 or later? Could you also include a picture of the converted yeast form?

A.R. The pictures of culture growth were taken after 20 days of culture, and on the same day microscopic examination was also performed. Unfortunately, we can’t include a picture because we didn’t take photos of the yeast form.

  • The antifungal testing methods mentioned (lines 340 – 341, CLSI M27) are actually for Candidatesting, not filamentous or dimorphic fungi. There are currently no standardized methods for susceptibility testing for the dimorphs and there are no breakpoints (see M38).

A.R. Dear reviewer, thank you for your suggestion. We have removed the results from antifungal testing methods. Moreover, we have reported your comment in the main text to clarify this paramount point and to favour further research in this field.

Minor comments:

Line 49: Clarify that histoplasmosis is the leading cause of endemic fungal respiratory infections.

A.R. We have modified the manuscript according to your suggestion.

Lines 59-60 “H. capsulatum is the leading cause of endemic fungal respiratory infections”.

Typically MALDI libraries as described by the “depth” of their library rather than their “width”

A.R. We have modified the term as suggested.

Line 123: Filmarray respiratory panel also includes bacterial targets.

A.R. We have added in the material and method section this statement.

Lines 166-171: Add the gene targeted by the P1 and P2 primers.

A.R.  We have modified and added this information in the result section.

DNA was extracted from positive cultures suspected to be H. capsulatum using MOL gen Universal Extraction Kit (ME 188830, Adaltis, Italy) following the manufacturer indications. Nucleic acid was amplified using primers P1 P2, that amplify the Large Subunit (LSU) variable region on the 28S gene [27], according to the following protocol: taq DNApolymerase 2x (pcrBio) 10 µL, primer mix P1 P2 [10 picomoles/µl], each 2 µl, extracted sample 1 µL, H2O 7 µL, and amplified with 9700 PCR systems (Applied Biosystems, Foster City, Ca). After the amplification, the samples were purified by QIAquick pcr purification kit Qiagen (Qiagen, Hilden, Germany) and sequenced using Brilliant DYE Terminators cycle sequencing kit v 1.1 (BRD1-100), NimaGen Lagelandseweg 56 6545 CG Nijmegen-The Netherlands. The sequences were analyzed by ABI Prism 310 (Applied Biosystems, Foster City, Ca). Primers P1 ATCAATAAGCGGAGGAAAAG and P2 CTCTGGCTTCACCCTAATC were used for both amplification and sequencing. The electropherogram was analyzed by "Sequence Analysis" Version 5 and the sequence was analyzed by the NCBI BLAST program.”

Add a scale bar to microscopic pictures in Figure 1 and change “yellow arrows” in description to “red”.

A.R. We have added a scale bar to microscopic pictures in Figure 1 and changed “yellow arrows” to “red” in the figure legend.

Line 228: reword as labs can either update their Bruker databases with the updated filamentous fungal library once it is approved, or they can construct and validate a Histoplasma library themselves as Valero et. al did.

A.R. Dear reviewer thank you for your suggestion. We have modified and implemented the main text in the section 4.3 as follows:

Thus, the laboratories can either implement Bruker databases with the updated filamentous fungal library once it is approved or construct and validate a Histoplasma library themselves as Valero et al. [28] did for a correct identification of the fungus.”

Lines 238-240: Be specific why 6 out of the 12 articles you found were excluded.

A.R. We have specified this point as follows:

Of these, four were excluded after the application of exclusion criteria (articles discussing proteomic methods other than MALDI-TOF and non-English language publications). Among the eight remaining publications eligible for analysis, two were excluded after reading the abstract or the full text (did not concern cases of identification of H. capsulatum by MALDI in human).

Line 326: Expand. Antibody testing can also cross react and it takes weeks for antibodies to arise so it’s not as helpful as a diagnostic.

A.R. Dear reviewer, thank you for your suggestion. We have implemented the introduction and discussion section as follows:

“Several factors influence the reliability and reproducibility of molecular tests, including the type of samples tested. Fungal culture represents the best-performing sample but this has little applicability in the routine diagnostic laboratory, since the fungus can take a long time 4 to 6 weeks to growth before PCR application [37]. The whole blood and serum are also the appropriate specimen for PCR testing, with a very high sensitivity and specificity, whereas molecular diagnosis can be applied to lung biopsies when other classical diagnostic methods used for Histoplasma identification are usually not effective [20]. Although many of the PCR tests for Histoplasma identification were developed in-house and a meta-analysis study showed the lack of consensus regarding the methodologies, particularly PCR protocols and gene targets (100-KDa-like-protein genes and amplification of ITS regions), molecular tests have been shown to have a good clinical performance in patients with histoplasmosis [20]. As reported by Martagon-Villamil et al. [38] PCR assay has shown to be 100% sensitive and 100% specific for the differentiation of H. capsulatum from other genetically related fungi, including B. dermatitidis, or with fungi that have forms that resemble those of C. glabrata, S. schenckii or T. marneffei. Most studies reported in the literature have been based on nested PCR, and more recently, real-time quantitative PCR (qPCR) methods have been developed for human diagnosis to overcome the limits of the microbiological culture [20]. To improve sensitivity of existing real-time quantitative PCR (qPCR) assays, Alanio et al. [4] further developed a new RT-qPCR assay that allows amplification of whole nucleic acids (WNA) instead of DNA and ribosomal small subunit RNA (mtSSU) instead of ITS as the target gene of Histoplasma spp.. This test performed better in the detection of H. capsulatum var. duboisii than the other conventional diagnostic methods [4]. This is of great clinical importance considering that histoplasmosis due to H. capsulatum var. duboisii is frequently underdiagnosed in Africa. Furthermore, a highly sensitive assay on blood can help with posttreatment follow-up of patients with histoplasmosis. Thus, to date, molecular approaches may be considered a promising laboratory tool to identify H. capsulatum from clinical specimens or cultures in all countries where histoplasmosis infection is not endemic.

Figure S1: Label each spectra and add the Histoplasma spectra as reference.

A.R. Dear reviewer, thank you for your suggestion. There are no reference spectra for Histoplasma capsulatum in the Bruker library, so we can’t add it to the text.  Moreover, the raw peaks reported in Figure S1 refer to the raw peaks found during the analysis of the four wells that were misidentified, as reported. Still, we cannot report the raw peaks of the reference strains included in the Bruker database as the raw peaks can be obtained only after MALDI analysis.

Figure S2: show the subject ID. Can include a screenshot of the blast results with the percent identification.

A.R. Dear reviewer, thank you for your suggestion. The subject ID cannot be shown, but a screenshot has been uploaded as material for the reviewers, not for publishing but only for the MALDI. The blast results with the percent identification were added in the capture of Figure S2.

Figure S3: Don’t use. This is for Candida, not Histoplasma and cannot be adapted.

A.R. Dear reviewer, thank you for your suggestion. We think you are referring to Table S1, the yeast MIC value, and not to Figure S3. According to your request, we have removed the Table S1.

Reviewer 2 Report

The manuscript “Closing the gap in proteomic identification of dimorphic fungi: 2 a case report of Histoplasma capsulatum infection and review of literature” analyzes the failure in the use of MALDI-TOF as a diagnostic method to identify Histoplasma capsulatum in a patient with HIV. It is an interesting article since the authors review the application of the MALDI-TOF method in diagnosing histoplasmosis. I also consider that implementing this diagnostic method for histoplasmosis in hospitals would be very useful. However, I have some comments.

Title

The title does not agree with the objective of the work since it mentions “Closing the gap in proteomic identification of dimorphic fungi” and only the proteomic identification of Histoplasma is analyzed.

Introduction

Line 41. The authors mention that there are three varieties of Histoplasma; however, I consider that this classification should no longer be used since H. capsulatum is currently considered a complex of cryptic species, consisting of several groups of isolates that differ genetically and they correlate with a particular geographic distribution. Histoplasma has been isolated on five continents, and currently, based on several genetic diversity studies, the phylogeography of the H. capsulatum complex has been reorganized into at least 14 phylogenetic species and four lone lineages distributed worldwide. Furthermore, Sepúlveda et al. published new adjustments to the taxonomy of H. capsulatum, who used a concatenated phylogenetic reconstruction in a whole-genome assembly study. They renamed four geographic groups of Histoplasma from the American continent, such as H. capsulatum sensu stricto Darling 1906 (instead of the H81 lineage from Panama); H. mississippense sp. nov. (instead of NAm 1); H. ohiense sp. nov. (instead of NAm 2); and H. suramericanum sp. nov. (instead of LAm A). Therefore, I consider that these classifications should be mentioned.

Materials and methods

Line 169. Mention the reference of the primers used for the PCR and justify using this marker.

Line 362, Italicize Histoplasma

Results

In the section, “4.3. Identification of H. capsulatum by Sequencing of PCR-amplified DNA”, the authors do not describe the procedure to conclude that the sequences obtained with the marker allow us to infer that it is Histoplasma capsulatum.

Discussion

The authors should expand the discussion about the usefulness of molecular methods for the identification of Histoplasma, since the current discussion is focused on the advantages and disadvantages of MALDI-TOF.

References

Unify the references according to the journal's instructions

Author Response

Dear Editor, 

Thank you very much for giving us the opportunity to revise our manuscript (Submission ID: 2595986). We also would like to thank the reviewers for their valuable comments that undoubtedly will improve the quality of our paper.

Here, we wrote a point-to-point letter in order to respond to the reviewer’s comments. All the changes made are in red font (mark copy).

The manuscript “Closing the gap in proteomic identification of dimorphic fungi: a case report of Histoplasma capsulatum infection and review of literature” analyzes the failure in the use of MALDI-TOF as a diagnostic method to identify Histoplasma capsulatum in a patient with HIV. It is an interesting article since the authors review the application of the MALDI-TOF method in diagnosing histoplasmosis. I also consider that implementing this diagnostic method for histoplasmosis in hospitals would be very useful. However, I have some comments.

Title

  • The title does not agree with the objective of the work since it mentions “Closing the gap in proteomic identification of dimorphic fungi” and only the proteomic identification of Histoplasmais analyzed.

A.R. Dear reviewer thank you for your suggestion. We have modified the title as follows: “Closing the gap in proteomic identification of Histoplasma capsulatum: a case report and review of literature”.

Introduction

  • Line 41. The authors mention that there are three varieties of Histoplasma; however, I consider that this classification should no longer be used since  capsulatumis currently considered a complex of cryptic species, consisting of several groups of isolates that differ genetically and they correlate with a particular geographic distribution. Histoplasma has been isolated on five continents, and currently, based on several genetic diversity studies, the phylogeography of the H. capsulatum complex has been reorganized into at least 14 phylogenetic species and four lone lineages distributed worldwide. Furthermore, Sepúlveda et al. published new adjustments to the taxonomy of H. capsulatum, who used a concatenated phylogenetic reconstruction in a whole-genome assembly study. They renamed four geographic groups of Histoplasma from the American continent, such as H. capsulatum sensu stricto Darling 1906 (instead of the H81 lineage from Panama); H. mississippense sp. nov. (instead of NAm 1); H. ohiense sp. nov. (instead of NAm 2); and H. suramericanum sp. nov. (instead of LAm A). Therefore, I consider that these classifications should be mentioned.

A.R. Dear reviewer, thank you for your clear and useful explanation about cryptic species. We agree with you and we have modified and implemented the introduction in the “epidemiology” section. We have also cited all the papers you suggested.

Materials and methods

  • Line 169. Mention the reference of the primers used for the PCR and justify using this marker.

A.R. Dear reviewer thank you for your suggestion. These primers amplify the nuclear large subunit (LSU) variable region on the 28S gene as reported by Sandhu et al.  [Sandhu GS, Kline BC, Stockman L, Roberts GD. Molecular probes for diagnosis of fungal infections [published correction appears in J Clin Microbiol 1996 May;34(5):1350]. J Clin Microbiol. 1995;33(11):2913-2919. doi:10.1128/jcm.33.11.2913-2919.1995].

We have included this statement in the results section:

To definitively confirm the laboratory diagnosis of histoplasmosis, the DNA from fungal culture was extracted and characterized by sequencing the LSU regions of the 28S gene. The results from sequencing analysis aligned on BLAST. (NCBI: https://blast.ncbi.nlm.nih.gov/Blast.cgi) (Figure S2) confirmed the culture suspected isolate as H. capsulatum var. capsulatum. The results of sequencing have been deposited at the GeneBank and have the following accession number OR604568.”

  • Line 362, Italicize Histoplasma

A.R. We have corrected and italicized the name.

Results

  • In the section, “4.3. Identification of  capsulatumby Sequencing of PCR-amplified DNA”, the authors do not describe the procedure to conclude that the sequences obtained with the marker allow us to infer that it is Histoplasma capsulatum.

A.R. Dear reviewer, thank you for your suggestion. We have added and clarified the sentence as suggested. The investigated gene is the 28S of Large Subunit (LSU).

We have included this statement in the results section as follows:

To definitively confirm the laboratory diagnosis of histoplasmosis, the DNA from fungal culture was extracted and characterized by sequencing the LSU regions of the 28S gene The results from sequencing analysis aligned on BLAST. (NCBI: https://blast.ncbi.nlm.nih.gov/Blast.cgi) (Figure S2) confirmed the culture suspected isolate as H. capsulatum var. capsulatum. The results of sequencing have been deposited at the GeneBank with the following accession number OR604568.”

Discussion

  • The authors should expand the discussion about the usefulness of molecular methods for the identification of Histoplasma, since the current discussion is focused on the advantages and disadvantages of MALDI-TOF.

A.R. Dear reviewer, thank you for your suggestion. We have implemented the discussion as requested to underline the usefulness of molecular methods for the identification of Histoplasma spp.

References

  • Unify the references according to the journal's instructions.

A.R. We have unified the references according to the journal's instructions.

Reviewer 3 Report

Dear EIC, Prof. Dr. David S. Perlin

Dear AE, Dr. Estelle Zhang

I hope you are doing well.

This is my review result for manuscript # jof-2595986.

The authors reported a rare case H. capsulatum. The methodology is acceptable and the literature is well reviewed. However, I prepare two comments as results of my review. Comment two is the major one.  

·      Introduction should be truncated to less than one page.

·      The results of sequencing should be deposited at the GeneBank and its accession number should be inserted at the section 4.3.

·       

Author Response

Dear Editor, 

Thank you very much for giving us the opportunity to revise our manuscript (Submission ID: 2595986).

We also would like to thank the reviewers for their valuable comments that undoubtedly will improve the quality of our paper.

Here, we wrote a point-to-point letter in order to respond to the reviewer’s comments. All the changes made are in red font (mark copy).

The authors reported a rare case H. capsulatum. The methodology is acceptable and the literature is well reviewed. However, I prepare two comments as results of my review. Comment two is the major one.

  • Introduction should be truncated to less than one page.

A.R. Dear reviewer, thank you for your suggestion. As requested by you and by Reviewer 1 we have modified the introduction and divided it in three sections (epidemiology, pathogenesis and diagnostic tools) to make the reading easier.

  • The results of sequencing should be deposited at the GeneBank and its accession number should be inserted at the section 4.3.

A.R. Dear reviewer, thank you for your suggestion. The results of sequencing were deposited at the GeneBank with the accession number reported in the section 4.3. We have added this information in the results paragraph.

The results of sequencing have been deposited at the GeneBank with the following accession number OR604568.

Round 2

Reviewer 3 Report

I hope you are doing well

I convinced by author feedbacks to my comments.